# Sex Differences in the Relationship between Student School Burnout and Problematic Internet Use among Adolescents

**DOI:** 10.3390/ijerph16214107

**Published:** 2019-10-24

**Authors:** Katarzyna Tomaszek, Agnieszka Muchacka-Cymerman

**Affiliations:** Department of Psychosomatic, Institute of Psychology, Pedagogical University of Cracow, Podchorążych 2, 30-084 Kraków, Poland; katarzyna.tomaszek@up.krakow.pl

**Keywords:** sex differences, school burnout, Internet addiction, adolescents

## Abstract

Background: The Internet has many positive sides, but it can also have a negative impact on human emotional-cognitive and behavioral functioning, especially during adolescence. To the most common consequences, the authors add addiction of the teenager to the Internet. This addiction is related to many negative physical and mental problems, including depression, substance abuse and social isolation. Methods: In the study, SSBS (Student School Burnout Scale) was used to measure the level of burnout, and the level of Internet addiction was measured using PUI (Internet Addiction Test). The research was carried out among 230 individuals aged 17–20 years. Results: The results of the research showed that higher level of school burnout is related to higher Internet addiction indicators, and connection was stronger in the male group. Gender significantly predicted Internet addiction and moderated the link between school burnout and Internet addiction. Hierarchical multiple regression analyses confirmed different predictors of Internet addiction for male and female students. However, for both groups, higher burnout due to parents was a significant predictor of Internet addiction (IA). Discussion: Internet addiction was predicted by school burnout, appearing as a result of parental pressure for high school achievements. In addition, school burnout and school-related characteristics have greater prediction power of Internet addiction and its indicators in a male group.

## 1. Introduction

Research on the problem of burnout is most often carried out on a group of employees representing social professions. There are several works and studies devoted to the causes, manifestations and effects of occupational burnout among these professions [1]. Among the social professions, one of the groups most vulnerable to burnout is the teacher. A burned out teacher has a negative impact on students, and the consequences of this influence cause the students’ burnout. Burnout among students is growing and concerns more students [2]. Epidemiological studies on school burnout in Finland revealed that 10–15% of adolescents suffered from this problem [3], and studies from other countries among medical students have estimated burnout to be 18–76% [4]. Over the past few decades, this construct started to be introduced as one of the main reasons of serious physical and psychological problems among youths such as: maladjusted behavior at school, low school performance and high risk of educational failure, psychosomatic and emotional disorders (depression), and alcohol and drug abuse [5,6]. Internet addiction (IA) is a mental health disorder, which is characterized by lack of patience, isolation and emotional disturbances and interruption of relationships with other people [7]. The results of research conducted in Europe showed that the IAD (pathological Internet use or problematic Internet use) prevalence among youths is 1–11% [8], and in the USA varies between 7.9% to 25.2% [9]. Even though Young created the IA measure by adapting the Diagnostic and Statistical Manual-IV (DSM IV) criteria, only online gambling disorder was added to the actual fifth version of this classification—DSM V [10]. Due to increase in the risk of Internet addiction and burnout among young people, and its negative outcomes such as emotional problems, depression, isolation of social relations and educational low performance, it seems to be important to study its relationship to student school burnout.

### 1.1. Student School Burnout

Student school burnout syndrome is a psychological problem caused by the continuous exposure on school related stress factors. Burnout can be defined as school related strain and stress, the reaction that emerged when students are challenging the pressure to fulfil their educational duties [3,11]. People experiencing burnout look and act as if they are depressed [12]. It is worthwhile to notice that some authors defined school burnout syndrome as the negative pole of continuous subjective well-being [13,14]. The most popular meaning of this construct refers to three-dimensional theory [15]. They described burnout as an emotional exhaustion (the central quality of burnout that refers to lack of energy, increased irritability, anhedony and the depletion of emotional resources), cynicism (also named depersonalisation, which refers to the process of withdrawal and disconnection, individual distance between oneself and others, and is manifested by indifference and negativity towards school responsibilities), and low personal accomplishment (also named inefficacy, which refers to the sensation of lack of competence in the performance of educational tasks and the sense of school failure) [15,16]. However, a universal definition of this construct is still discussed up to the present [17]. Some conceptualizations of this phenomenon distinguish only one dimension, that of exhaustion [18,19]. Some authors [14] revealed a 7-factor structure of adolescent burnout: Loss of interest in school, burnout due to studying, burnout due to family, burnout due to doing homework, lethargy and boredom due to teacher attitudes, need for relaxation and entertainment, and deficiencies in the school. Despite the disagreements about the number of burnout dimensions, researchers agree that student school burnout makes students feel unable to effectively and successfully fulfill their school responsibilities. The most common cause of burnout among young people is the enormous pressure on the student resulting in the feeling of school stress. School burnout is not only related to difficulties in school performance, such as low school engagement and diminishing academic and cognitive performance, but also other factors, such as demotivation, low self-esteem and decreased self-efficacy [20,21], anxiety, negative affect and depression [22,23], alienation, withdrawal and isolation [24], the negative perceptions about social environment [3,25,26], ineffective coping strategies, and external locus of control [27]. Research carried out on a group of primary school students in Poland for the factors contributing to burnout additionally includes exhaustion, which is related to parental pressure, while high school students are the most deprived of school activities [28]. The exhaustion experienced by students results in a lack of self-control, poor resistance to stress, reduced effectiveness of actions, etc. Due to the decrease in motivation to take up further educational challenges, teenagers are escaping into various forms of activity that can lead to addiction. School burnout is higher among boys than girls [29,30,31]. 

### 1.2. Internet Addiction among Adolescents

Internet addiction (IA) (and related terms such as Problematic Internet Use, Excessive Internet Use, Pathological Internet Use (PIU), Internet Dependence, and Technology Addiction), is a type of behavioral addiction that involves the interaction between humans and machines [32,33]. Researchers define this construct as a lack of ability to control one’s use of the Internet which persists over a significant period of time, even though the variety of negative consequences of this activity affect the user’s life [9,34]. Additionally, characteristics of Internet-addicted students usually include alcohol and drug abuse, smoking, online and offline antisocial behavior, health problems, individuals’ suicidal ideation and depressive symptoms, withdrawal and poor school performance, as well as attention deficit and hyperactivity [34,35,36]. Several diagnostic criteria for Internet addiction have been proposed, which are: preoccupation, mood change when attempting to stop Internet usage, the need to use the Internet for increasing amounts of time, unsuccessful efforts to stop using the Internet, staying online longer than intended, lying about Internet use, neglecting relationship or opportunities, escape from problems or seeking to relieve bad mood states [37]. Mentally immature young people are considered to be the potential risk group for IA [38]. Unlike other addictions, IA is observed in earlier ages [39]. The increased popularity of social networks, online games, and the variety of Internet-based applications that are easy for adolescents to obtain and use, often leading them to use them day and night [10]. According to many researchers, these characteristics of the Internet increased the risk of an uncontrollable urge to use it and being obsessed with the Internet [10]. What is more, obsessive use of the Internet leads to IA, that has a negative impact on the mental health of adolescents [40]. Some researchers report that 1.4 to 17.1 percent of young people across the globe are addicted to the Internet [41]. The higher Internet addiction level was found in a group of 19 years and below young people in comparison to individuals aged 20 years and above [42]. Gender differences in adolescents’ internet addiction had also been reported. The previous findings showed that men are 50% more likely to be addicted to the Internet than women. Men more often become addicted to online games [43]. A meta-analysis that involved 39 global jurisdictions revealed an average but significant effect in the size of gender-related differences in IA [44]. Heavier addictions correlate with lower self-esteem and lower life satisfaction [43]. 

### 1.3. Student School Burnout and Internet Addiction

Following the previous findings, burnout and Internet addiction are considered to be a serious mental health problem that is related to chronic stress, and is most common among adolescents. Both are related to multiple negative phenomenon that affects educational performance (lower academic performance and academic failure), social relationships (lack of social support offline and social deficits experienced), and physical and psychological functioning [45,46,47]. Burnout is a negative emotional reaction which occurs when resources are threatened, lost, or unavailable for people in stressful situations [48]. What is interesting is that some researchers claim that Internet use may be viewed as a maladaptive coping response to emotional or social difficulties which also occur as a result of a lack of appropriate coping resources [47,49]. However, psychopathological characteristics such as depression and social anxiety, personality traits such as self-esteem, self-efficacy, and stress vulnerability, and social cognitions such as emotional loneliness and perceived social support, impact symptoms of Internet addiction only indirectly [47]. The main influence of these variables was mediated by a dysfunctional coping style, and Internet use expectancies. Stress was found to be a strong predictor of Internet addiction [50], and Internet addiction predicted stress [51], and negative quality of life [52]. Finally, along these lines, the two most popular theories within burnout research, Jobs Demands Resources Model (JD-R) and the Conservation of Resources Model, posit that burnout predominantly occurs and is strongly associated with untreated or unresolved chronic stress, tension, and distress [53,54]. We cannot overlook that some of the burnout symptoms appear to resemble the ones of Internet addiction. Numerous studies have reported strong connections of school burnout and Internet addiction with depression, anxiety, psychological distress, and stressful life events in adolescents and students [22,23,55,56,57]. Both constructs are also associated with social withdrawal, isolation, and loneliness [24,58,59]. Additionally, risky behaviors and health problems have been linked to school burnout and Internet addiction such as: alcohol and drug abuse, delinquency, and suicide ideation [34,35,60,61,62,63,64]. So far, only a few studies examined the association between student burnout and Internet addiction. Problematic Internet use increases the risk of developing burnout among adolescents. What is more, in turn, school burnout may lead to digital addiction [65]. From the findings, gender played an important role in this relationship e.g., females suffered more than boys from school burnout symptoms in late adolescence, and males suffered from excessive Internet use. Internet addiction explained 16% of variances in the level of academic burnout [66]. The findings also showed that more often Internet addiction affects boys than girls, but there are no significant gender differences associated with school burnout. It is worthwhile to notice that student burnout is also associated with other technological addictive behaviors such as mobile phone dependency, which explained 47.6 of the variances on academic burnout [67]. This research was conducted among undergraduate students at the School of Nursing and Midwifery. Considering the connections mentioned above, regarding the importance of school burnout and Internet addiction and its potential harmful effect on the development of youths, it is important to examine the associations between both of them. 

The literature seems to suggest that male college students are more prone to develop Internet addiction [68]. In addition, no study about the relationship between Internet addiction and school burnout could be traced among high school Polish adolescents. The aim of this study was to estimate sex differences of Internet addiction predictors. In particular, we hypothesised that Internet addiction is positively correlated to student school burnout indicators, and that in males these correlations are stronger. We also supposed that gender is a significant predictor of Internet addiction and that it moderates the link between school burnout and Internet addiction. Finally, we hypothesized that high school students with burnout syndrome, with poor quality relationships in the family and school environment, and with problems related to school performance, are more prone to develop Internet addiction.

## 2. Materials and Methods

### 2.1. Study Population and Data Collection 

The sample consisted of 230 individuals aged 17–20 years (M = 18.35 years, SD = 0.45). 74.8% of the participants were 18 years old (N = 172), 24.3% of the students were 19 years old (N = 56), 0.45 % of the participant students were aged 17 years old (N = 1), and also 0.45% were 20 years old students (N = 1). The study group included 171 females (74% of the sample), and 59 males (26% of the sample). The participants were recruited in one type of school: senior high schools (Polish lyceum) from different regions in Poland. The participants volunteered for the study; they were approached in their classrooms and asked to fill in the methods, and informed about the anonymity of the research. They received no reward for their participation in the study.

### 2.2. Measures

The Student School Burnout Scale (SSBS) by Ayşe Aypay was used [14]. Student school burnout is a phenomenon ussualy measured by methods adapted from instruments that were originally intended for workers. However, research carried out on students should take into account the specificity of the developmental period (immaturity of certain mental structures), and the much greater importance of family and peer environments for coping with stressful school situations. What is more, the performance of professional work is not the same as the obligation to learn at school, especially because of the legal obligation of students to implement education in the education system. For that reason, Ayşe Aypay developed the inventories that directly measure student burnout in school contexts [69]. We decided to use one of these inventories because in our opinion, it enables in-depth analysis of school burnout and it is tailored to the complex situation of young people. Additionally, it is created on the basis of the problems identified by the students themselves. The SSBS scale includes 34 items categorised into seven sub-scales listed as: LIS—Loss of Interest in School (example question: *I don’t want to go to school*), BDS—Burnout Due to Studying (s.q.: *I think studying is meaningless*), BDF—Burnout Due to Parents (s.q.: *I’m often scolded by my parents because of the problems I have at school*), BDH—Burnout Due to Doing Homework (s.q.: *I’m so bored with doing homework that I always postpone doing it*), BTT—Being Bored and Tired of Teacher Attitudes (s.q.: *I cannot stand my teachers’ overcontrolling actions*), NRF—Need to Rest and Have Fun (s.q.: *I can’t find the time for resting and having fun because I have a lot of homework*), ISS—Incompetence in School (s.q.: *I often feel incompetent while doing my homework*). All items were scored on a 4-point Likert scale where the examined person marks 1—if strongly agrees, to 4 if strongly disagrees. When the students rated how strongly they agree/disagree with the statement, their answer was marked by an X in the appropriate box. The lower the SSBS score, the higher the burnout result for that student. The reliability from pilot studies was satisfying for most of the subscales. Cronbach’s Alfa for the total score was ranged between α = 0.76–89, for subscales: LIS α = 0.74–84, BDS α = 0.38–77, BDF α = 0.67–79, BDH α = 0.66–67, BDT α = 0.28–58, NHF α = 0.65–77, ISS α = 0.72–76.

The Internet Addiction Test (IAT) is based on the Internet Addiction Diagnostic Questionnaire- IADQ [70]. The Polish version of this scale was named Problematic Use of the Internet (PUI) [71]. Originally, the IA test consists of 20 self-reported items scored on a 5-point Likert-type scale ranging from “rarely” to “always”. In the Polish version, there are 22 questions scored on a 6-point Likert-scale. Four-factor structure of the IAT test was revealed [72], with four dimensions named: Lack of Control (s.q.: *How often do you realize that you have been online for longer time than you originally intended*?), Social Withdrawal and Emotional Conflict (s.q.: *How often being online allows you to ease your negative feelings (e.g. hopelessness, sadness, depression, anxiety or guilt?)*, Time Management Problems (s.q.: *How often do people around you complain about the amount of time you spend online?),* and Concealing Problematic Behaviour (s.q.: *How often do you try to hide how long you stay online?*). This solution was tested and confirmed [73]. In their work, they distinguished such dimensions of Internet addiction as: Emotional and Cognitive Dependence (ECD), Time Management Problems (TMP), Lack of Control and Neglecting of Social Life, and Dishonesty about Internet Use (DI). The reliability estimates by Cronbach’s Alfa for IAT Total Score was equal to α = 0.90, and for the sub-scales were α = 0.87 (ECD), α = 0.84 (TMP), α = 0.74 (CSL), and α = 0.59 (DI), respectively.

The School-Related and Socio Demographic Characteristics Questionnaire was where high school students took part in the study anonymously. For the needs of the study, information was collected from the participants using the Personnel Questionnaire. The survey was used to collect information such as: gender, age, subjective assessment of completion of the role of the student, the time devoted to learning during the day, the average of the grades from the half-year, participation in congresses/olympiads, attending private tutoring, and the quality of relationships in the family.

### 2.3. Statistical Analysis

Pearson’s coefficient was calculated using STATISTICA 13.3 PL. Next, the SPSS version 22.0 (IBM, IBM House, Shelbourne Road, Ballsbridge, Dublin, Ireland) with PROCESS macro was used to estimate regression model and perform bootstrap method for testing moderation effect of gender on the link between SSBS and PUI. Finally, the authors calculated several multiple regression models to find significant predictors of PUI indicators. Ethical approval was issued by the Ethics Committee of the Pedagogical University of Cracow (WP.113-6/2019).

## 3. Results

The bivariate Pearson’s correlations between the IAT indicators and the SSBS scores in the scales administered among adolescents are shown separately for girls and boys in Table 1.

According to the analysis concerning the data of male students, there were significant negative correlations among all PUI indicators and the SSBS total score (coefficients ranged from r = 0.27, *p* < 0.05 to −0.52, *p* < 0.0001). Almost all SSBS sub-scales significantly correlated with ECD (except BTT and NRF scales) (coefficients ranged from r = −0.32, *p* < 0.05 to −0.60, *p* < 0.0001). TMP was significantly associated with BDF, NRF and ISS scales (coefficients ranged from r = −0.31, *p* < 0.05 to −0.36, *p* < 0.01). Higher level of CSL was connected with higher burnout in the LIS, BDS and BDH dimensions (coefficients ranged from r = −0.31, *p* < 0.05 to −0.38, *p* < 0.01). DI was significantly associated with BDF and ISS (r = −0.33, *p* < 0.05, and r = −0.33, *p* < 0.05, respectively). 

In the group of girls, Pearson’s correlation procedures showed statistically significant positive relationships among almost all IAT indicators (except DI) and BDS (coefficients ranged from r = −0.17, *p* < 0.05, and r = −0.25, *p* < 0.01), and between TMP, DI and IAT and BDF (coefficients ranged from r = −0.17, *p* < 0.05, and r = −0.22, *p* < 0.01). Only TMP correlated significantly with SSBS (r = −0.16, *p* < 0.05).

In summary, female high school students tend to have less school burnout characteristics related to Internet addiction than their male equivalents, and they score lower than male students in terms of the significant correlation coefficients of Internet addiction (see Table 1).

### 3.1. The Moderation Effect of Gender on the Relationship between Student School Burnout and Internet Addiction

To test the hypothesis that Internet addiction problems are a function of student school burnout and gender, and more specifically whether gender moderates the relationship between school burnout and Internet addiction, a hierarchical multiple regression analysis was conducted. In the first step, two variables were included, those of school burnout total score and gender. These variables accounted for a 6% of variance in Internet addiction problems, (∆R^2^ = 0.06, F_(2, 226)_ = 6.729, *p* = 0.001) (see Table 2).

Next, the interaction term between school burnout and gender was performed with Bootstrap method. Examination of the interaction showed a significant partial moderation effect with B = −0.57, CI 95% (−0.90; −0.24), statistics for interaction mode R^2^ change = 0.05, F _(2, 226)_ = 11.754, *p* = 0.0007) (see Table 3). 

### 3.2. Predictors of Internet Addiction Indicators for Male and Female Samples—Multiple Regression Analysis

Hierarchical multiple regression analyses were performed to check if gender makes a difference in school burnout predictors of Internet addiction. 

The results of the regression analysis conducted for the female group in the second analysis showed that five explanatory variables were significantly associated with Internet addiction: BDS, BDF, LIS, Additional classroom lessons and Quality of teacher-student relationships. Parameters of the regression model: ∆R^2^ = 0.16, F = 7.44, *p* ≤ 0.0001 (see Table 4). In the male group, three variables significantly predicted Internet Addiction: BDF, BDH and Quality of classmates relationships (∆R^2^ = 0.45, F = 16.99, *p* ≤ 0.0001) (see Table 5).

## 4. Discussion

Many studies show the problem of burnout and its negative impact on some forms of addiction [74]. We found that if a lack of resources appear, an individual becomes burned out, and thus, the risk of addiction increases. Our assumption is consistent with other studies in which burnout may cause many behavioral addictions, such as Internet Addiction [75,76], substance abuse [77] or alcohol dependence [78]. Additionally, similar to our results, many studies have revealed that the risk factors of IA stem from feelings of a lack of social resources, and is manifested by poor social relationships, loneliness, lack of social skills, and lack of social support [79,80]. Following the Job Demands Resources Model chronic stress, which is sparked by excessive demands and lacking resources may—via burnout—cause negative outcomes such as poor performance and health [81]. What is more, by increasing resources, such as social support, we reduce the risk of burnout syndrome and health problems. One of these health problems are addictive behaviours, such as problematic Internet use. Academic burnout heightened the potential risk of addiction among adolescents [44]. The authors suggested that symptoms of school burnout, frustration, and a sense of disappointment in people cause maladaptive behaviours such as isolation adaptations, and addiction to alcohol and drugs. These are one of possible ways to escape from the present difficult situation and reduce the tensions.

The main objective of this research was to test for gender differences in school-related predictors of Internet addiction. Our study supplemented the existing literature by examining the association between school burnout and Internet addiction for adolescent girls and boys separately [66]. According to the results, it was discovered that a higher level of school burnout is related to higher Internet addiction indicators. However, in the female adolescent group these relationships were weaker (we noted a smaller number of significant Pearson’s correlations, all significant correlations were low). In the male adolescent group, we found a larger number of significant correlation coefficients and with higher effect size (low to moderate). In other words, young burnout boys are at a higher risk of suffering from Internet addiction than girls. This result is consistent with the statement that gender-related differences exist in most addictive behaviors [82]. Other research showed differentiated effects on Internet addiction by gender in such prevalence of IA like: school grades, parental education, alcohol drinking, smoking consumption, and substance use [83]. Many previous studies showed that Internet addiction affects boys more often [66]. One possible explanation is that they try to cope with negative school pressure and stress, and negative feelings and thoughts connected with burnout syndrome, by indulging themselves on the Internet, which is rather an avoidance and ineffective strategy of problem solving.

Secondly, gender significantly predicted Internet addiction, and moderated the link between school burnout and Internet addiction. Hierarchical multiple regression analyses confirmed different predictors of Internet addiction for female and male students.

Our findings suggest that school burnout and school-related characteristics have greater prediction power of Internet addiction in a male group than in a female group. As the results show, burnout due to parents appeared to be a significant predictor of IA for both male and female high school students. However, in the female group also, burnout due to study and loss of interest were significant predictors of PUI; in the male group only, burnout due to homework was significantly associated with PUI. What is more, in the female adolescent group, quality of teacher-student relationships had a significant connection to Internet addiction; in the male adolescent group it was quality of classmates’ relationship. It is well confirmed that parental pressure on school achievements lead to stress and anxiety in students [84]. Other studies prove that male students are pressurized by parents more than female students, however, female students’ levels of stress and anxiety were higher [85]. What is more, recent research has confirmed different family related factors of IA such as family support, family conflict, family functionality and approaches [10]. Perceived parental psychological control explained 18% of the IA level among students aged 14–18 [86]. In addition, peer pressure and perceived teacher support also were found to be a significantly related to IA [39]. There is an agreement between researchers and findings about the crucial role of deficits in social resources in the developing of IA [87]. Lack of “face to face” interactions with others may lead young people to look for them on the Internet [87]. Numerous studies have demonstrated the link between loneliness and IA [88]. The results of studies conducted among 300 high school students (M = 16.49) confirmed that perceived social support from significant others, loneliness, cognitive avoidance, and problem solving were among coping strategies which significantly predicted IA [89]. 

Many studies indicate a gender difference in patterns and features of IA [90]. Psychiatric distress and impulsivity may lead to problematic online gaming (POG), and this connection is stronger in males [83]. Impatience mediated the link between psychiatric distress and POG in the male group, whereas the indirect effect of impatience was insignificant in females. However, gender did not predict the IA level [38]. The mechanisms that lay under the sex differences in the level of IA is still unclear but may be connected to a different coping style in response to adversity. Female students reported a higher emotional reaction to stressors, and male students reported higher behavioral and cognitive reactions to stressors [91]. Women also scored significantly higher in emotional and avoidance coping styles as compared to men. However, gender was a significant predictor of nomophobia (irrational fear and anxiety arising from the feeling of disconnection from virtual communication platforms) and it was the male gender in the nomophobic group that was a predictor for avoidant coping style [92]. The findings suggest, that in a different type of digital addiction, the relations between gender, stress (and burnout sparked by it), and coping styles may differ. 

### Limitation and Strength

Despite this, we consider the results to be encouraging and compatible to previous studies. This study faced several limitations such as relatively small sample size, so the generalizability of results of this study decreases. For that reason, it is recommended to examine larger populations, to more clearly identify factors associated to school burnout and Internet addiction. Secondly, our study used a cross-sectional design, which excludes the possibility of drawing conclusions of a causal nature about the relationship between school burnout and Internet addiction. Another limitation was the fact that the information about the school environment relationships between burned out adolescents, their classmates and teachers, were collected by using only self-report, and adolescents’ subjective perspective. Further studies should include more objective perspectives, not only from the school environment, but also from out of school relationships. What is more, we did not control such demographic characteristics as place of living, family’s structure, family’s income level etc. Additionally, the study did not involve the information about other risky behaviors such as smoking, alcohol or drug abuse, and other Internet behaviors such as social networking, sharing, banking, shopping etc., or information about other technological addiction such as addiction from mobile phone [93]. In addition, some of the common information reported by other studies on Internet addiction were not reported, which could be related to Internet addiction and burnout. For example, frequency of online gaming [94] and money spent [95], information on computer ownership [96], places for Internet access [97], living arrangement and parental supervision [98], personality [99], strategies to cope with stress [100], and sleep quality [101]. This can have a significant impact on the results they obtained in the study. Additionally, further assessment of the validity of the SSBS is necessary because of the low reliability of two sub-scales: BDS and BDT. Our results can be useful for educational workers and planners, addiction therapists and policy makers, because students with a high risk for school burnout have high vulnerability for Internet addiction and vice versa. In other words, school burnout may be an early indicator of problematic Internet use. Our results also cannot exclude that people who are affected by Internet addiction have answered the questions in a meaningful manner. The moderation effect of gender on the relationship between SSBS and IA indicate the necessity of developing other ways to prevent IA for girls and boys. What is more, in preventive actions more emphasis should be placed on building a positive climate at school. Finally, since there was a difference between males and females in predictors of Internet overuse, the association between student burnout and IA should be examined in detail by gender with the use of a longitudinal study.

## 5. Conclusions

Our research confirmed that all kinds of strategies to help and prevent Internet addiction should also be aimed at minimizing the sense of burnout from the role of the student. The authorities must pay attention to the risk of students misusing the Internet and encourage them to limit the use of this media to prevent burnout in the future [66]. Activities related to the prevention of Internet addiction and school burnout should also focus on both the school and family environment. Likewise, any activities used to counteract addiction should be focused on the student and consistently followed by both the school and family environment. It is important to control the coping strategies of young people. Research shows that the loss of any coping strategies is the first step to the appearance of burnout in the individual and engaging in addictive activities such as Internet addiction.

## Figures and Tables

**Table 1 ijerph-16-04107-t001:** Sex differences of the correlation coefficients of Problematic Internet Use and Student School Burnout (N_female_ = 170, N_male_ = 59).

Variables	1	2	3	4	5	6	7	8	9	10	11	12	13
1.LIS	-	0.60 ***	0.06	0.49 ***	0.23	0.26 **	0.44 ***	0.74 ***	−0.32 *	0.04	−0.37 **	−0.15	−0.25
2.BDS	0.34 ***	-	0.22	0.53 ***	0.23	0.41 **	0.34 **	0.78 ***	−0.38 **	−0.20	−0.38 **	−0.09	−0.39 **
3.BDF	0.27 ***	0.12	-	0.09	−0.14	0.03	0.37 **	0.38 **	−0.60 ***	−0.32*	−0.25	−0.33 *	−0.55 ***
4.BDH	0.38 ***	0.34 ***	0.27 ***	-	0.24	0.46 ***	0.06	0.67 ***	−0.38 **	−0.07	−0.31*	−0.18	−0.31 *
5.BTT	0.46 ***	0.26 **	0.32 ***	0.28 ***	-	0.15	0.15	0.51 ***	−0.01	−0.11	−0.00	−0.06	−0.07
6.NRF	0.40 ***	0.20 **	0.13	0.33 ***	0.41 ***	-	0.04	0.52 ***	−0.12	−0.32*	−0.04	0.05	−0.22
7.ISS	0.31***	0.19 *	0.22 **	0.19 *	0.33 ***	0.53 ***	-	0.57 ***	−0.32 **	−0.36**	−0.24	−0.33 *	−0.43 **
8.SSBS	0.75 ***	0.57 ***	0.56 ***	0.63 ***	0.66 ***	0.65 ***	0.59 ***	-	−0.50 ***	−0.31*	−0.37 **	−0.27 *	−0.52 ***
9.ECD	0.13	−0.21**	−0.08	−0.03	0.00	0.12	−0.05	−0.03	-	0.32 **	0.57 ***	0.56 ***	0.86 ***
10.TMP	0.02	−0.22 **	−0.22 **	−0.07	−0.02	−0.06	−0.10	−0.16 *	0.43 ***	-	0.14	0.27 *	0.72 ***
11.CSL	0.14	−0.17 *	−0.03	0.02	0.06	0.11	0.05	0.04	0.72 ***	0.40 ***	-	0.38 *	0.63 ***
12.DI	0.06	−0.12	−0.19 *	−0.01	−0.03	0.13	−0.05	−0.05	0.40 ***	0.47 ***	0.43 ***	-	0.63 ***
13.IAT	0.10	−0.25 **	−0.17 *	−0.04	0.00	0.07	−0.06	−0.08	0.84 ***	0.80 ***	0.79 ***	0.63 ***	-

Note: *** *p* < 0.001; ** *p* < 0.01; * *p* < 0.05; In SSBS scale the higher score means the lower school burnout. LIS—Loss of Interest in School; BDS—Burnout Due to Studying; BDF—Burnout Due to Parents; BDH—Burnout Due to Doing Homework; BTT—Being Bored and Tired of Teacher Attitudes; NRF—Need to Rest and Have Fun; ISS—Incompetence in School; SSBS—Student school burnout; ECD—Emotional and Cognitive Dependence; TMP—Time Management Problems; CSL—Lack of Control and Neglecting of Social Life; DI—Dishonesty about Internet Use; IAT—Problematic Internet Use.

**Table 2 ijerph-16-04107-t002:** Multiple linear regression model of Problematic Internet Use.

Variables	R^2^	R^2^ Adjusted	R Change	B	SE	*Β*	t	*p*
SSBS	0.06	0.05	0.06	−0.27	0.08	−0.23	−3.445	0.0001
Gender				5.30	4.42	0.15	2.191	0.029
Model F = 6.729, *p* = 0.001								

Note: SSBS—Student school burnout.

**Table 3 ijerph-16-04107-t003:** Simple moderation effect of gender on the link between School Burnout and Internet Addiction.

Variables	Outcome				
	Coefficient	SE	t	*p*	CI 95%
SSBS	−0.25	0.08	−3.19	0.002	(−0.40; −0.09)
Gender	7.52	2.45	3.07	0.002	(2.69; 12.35)
SSBS x Gender	−0.57	0.17	−3.43	0.0007	(−0.90; −0.24)
Model Summary	F = 8.618	*p* ≤ 0.0001	R = 0.32,	R^2^ = 0.10	R^2^ change = 0.05

Note: SSBS—student school burnout total score. Percentile bootstrap CI based on 5000 bootstrap samples.

**Table 4 ijerph-16-04107-t004:** Multiple linear regression model of Problematic Internet Use for female group (N = 170).

Variables	R^2^	R^2^ Adjusted	R Change	B	SE	*β*	t	*p*
BDS	0.06	0.06	0.60	−1.15	0.35	−0.25	−3.282	0.001
Model F				10.774 ***				
BDS	0.10	0.09	0.39	−1.49	0.37	−0.32	−4.059	0.000
LIS				0.836	0.31	0.21	2.687	0.008
Model F				9.195 ***				
BDS	0.14	0.12	0.38	−1.486	0.37	−0.32	−4.059	0.000
LIS				0.836	0.31	0.21	2.687	0.008
BDF				−1.486	0.37	−0.32	−4.059	0.000
Model F				8.784 ***				
BDS	0.17	0.15	0.28	−1.328	0.36	−0.28	−3.695	0.000
LIS				0.972	0.31	0.24	3.107	0.002
BDF				−0.892	0.32	−0.21	−2.811	0.006
Additional classroom lessons ^i^				5.110	2.18	0.17	2.345	0.020
Model F				8.141***				
BDS	0.19	0.16	0.20	−1.245	0.36	−0.27	−3.473	0.001
BDF				1.110	0.32	0.28	3.497	0.001
LIS				−0.881	0.31	−0.21	−2.802	0.006
Additional classroom lessons ^i^				4.801	2.17	0.16	2.218	0.028
Quality of teacher-student relationships ^ii^				4.771	2.37	0.15	2.009	0.046
Model F				7.440 ***				

Note: *** *p* < 0.001; ^i^ 1—Yes I’m taking an additional classroom, 2—No, I am not taking additional lessons; ^ii^ 1—very close, 2—satisfactory, 3—very bad; *F:* Test of equality of variances; R^2^: Explained variance; B: Unstandardized regression coefficient; SE: Standard Error; β: Standardized regression coefficient; t: t-value. BDS—Burnout Due to Studying; BDF—Burnout Due to Parents; LIS—Loss of Interest in School.

**Table 5 ijerph-16-04107-t005:** Multiple linear regression model of Problematic Internet Use for male group (N = 59).

Variables	R^2^	R^2^ Adjusted	R Change	B	SE	*β*	t	*p*
BDF	0.30	0.29	0.30	−2.704	0.54	−0.55	−4.985	0.000
Model F				24.847 ***				
BDF	0.41	0.39	0.11	−2.582	0.50	−0.53	−5.126	0.000
Quality of classmates relationships ^i^				10.149	3.14	0.33	3.237	0.002
Model F				19.729 ***				
BDF	0.48	0.45	0.07	−2.466	0.48	−0.50	−5.138	0.000
Quality of classmates relationships ^i^				10.090	2.98	0.33	3.391	0.001
BDH				−1.601	0.60	−0.26	−2.677	0.010
Model F				16.990 ***				

Note: *** *p* < 0.001; F: Test of equality of variances; R^2^: Explained variance; B: Unstandardized regression coefficient; SE: Standard Error; β: Standardized regression coefficient; t: t-value. ^i^ 1—very close, 2—satisfactory, 3—very bad; BDF—Burnout Due to Parents; BDH—Burnout Due to Homework.

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
