# Peer review of "Sex Differences in the Relationship between Student School Burnout and Problematic Internet Use among Adolescents"

_ijerph, 2019, doi:10.3390/ijerph16214107_

Round 1

Reviewer 1 Report

The sentence in the Methods section f the Abstract is not complete "At the beginning, the students responded to a short survey".. There is supposed to be a follow up sentence. In the Discussion of the abstract, the author concluded that burnout is due to parental pressure. But this finding was never mentioned in the Results section of the Abstract.

In regression analysis, the authors found five explanatory variables that significantly associated with internet addiction use - BDS, BDF, LIS, classrooms lessons and Quality of teacher classroom relationship. There are not being discussed in the Discussion of the manuscript.

Author Response

Thank you for all supportive and constructive comments, and your time spending to check our text

We addressed the comments as follows:

The sentence in the Methods section f the Abstract is not complete "At the beginning, the students responded to a short survey".. There is supposed to be a follow up sentence.

Thank you for this comment, and we have decided to remove this sentence from the abstract, because it repeated the information about the methods used during the data collection.

In the Discussion of the abstract, the author concluded that burnout is due to parental pressure. But this finding was never mentioned in the Results section of the Abstract.

Thank you. We have added the information about the burnout due to parental pressure in the results in the abstract section.

In regression analysis, the authors found five explanatory variables that significantly associated with internet addiction use - BDS, BDF, LIS, classrooms lessons and Quality of teacher classroom relationship. There are not being discussed in the Discussion of the manuscript.

Thank you for very constructive comment. We admit, that we have actually skipped these issues so we have added discussion in the proper section

Reviewer 2 Report

I read this manuscript with interest and I think it can be considered for publication, should the authors be prepared to address some issues.

One of the main points is that the sample is composed of young adults, rather than adolescents. This fact could affect both the description of the background literature and the conclusions drawn from the data. I suggest stressing this point at least in the limitation section. In the abstract, the authors use the term "psyche". I think it could be more prudent and appropriate to use the term emotional-cognitive and behavioral functioning. Moreover, the authors state that "this addiction has many consequences..". However, there is an ongoing debate whether IA is consequence or predictor of other psychological difficulties. It is not clear why the authors state that the results have been obtained "controlling for gender", when the paper seems to address exactly the gender differences. In the introduction, whether or not IA is included in the DSM is not clear. At the first two lines of page 2, why "opportunities"? It is not clear why adolescence should be considered a particularly  at-risk period for IA. Why the authors think that sex differences are important in this field? I suggest adding a sub-heading "Measures". The authors should clarify why they choose these measures over others and give item examples for each measure. One, major, problem is that the paper would greatly benefit from professional English editing. A part from the many grammar mistakes, I think that the message of the paper could be better conveyed by revising the text fixing errors and above all amending the tone and word choosing.

Author Response

Thank you for your work and interest, and all constructive suggestions.

We addressed the comments as follows:

One of the main points is that the sample is composed of young adults, rather than adolescents. This fact could affect both the description of the background literature and the conclusions drawn from the data. I suggest stressing this point at least in the limitation section.

We cannot agree with this statement because of two reasons.

Firstly, the analysis of original 1482 definition of adolescence referred to a period between childhood and adulthood that extended between ages 14 and 25 years in males and 12 and 21 years in females (Oxford English Dictionary). The American Academy of Pediatrics “Bright Futures” defines adolescence as the ages of 11-21 years (AAP, 2015), and the World Health Organization name “adolescents” as individuals between 10 and 19 years, “youth” between 15 and 24 years, and “young people” between 10 and 24 years (WHO, 2015) (References: Curtis, A. M. (2015). Defining adolescence. Journal of Adolescent and Family Health. 7(2), Article 2)

Secondly, all sample consisted undergraduate high school students aged between 17 to 20. However most of participants were aged 18 (N = 172) or 19 years old (N = 56), only one student was older – 20 years old.

However, this comment made us aware of the need to refine the description of the group. For that reason we added information on the age structure of participants in the Study population and Data Collection section.

In the abstract, the authors use the term "psyche". I think it could be more prudent and appropriate to use the term emotional-cognitive and behavioral functioning.

Thank you, we have removed this statement  from the article.

Moreover, the authors state that "this addiction has many consequences..". However, there is an ongoing debate whether IA is consequence or predictor of other psychological difficulties.

We agree that this statement is too relative, so we have changed it in the whole article

It is not clear why the authors state that the results have been obtained "controlling for gender", when the paper seems to address exactly the gender differences.

Thank you, and we have changed this statement. 

In the introduction, whether or not IA is included in the DSM is not clear.

We added the information about the DSM classification in introduction

At the first two lines of page 2, why "opportunities"?

It was our mistake; we have changed this for a more appropriate term

It is not clear

why adolescence should be considered a particularly  at-risk period for IA. Why the authors think that sex differences are important in this field?

Fallowing your suggestion, we added this information in the “Internet addiction among adolescents” section

I suggest adding a sub-heading "Measures". The authors should clarify why they choose these measures over others and give item examples for each measure

Thank you, this comment is very constructive, we have added the measures section, included proper information and sample questions to each subscale.  

One, major, problem is that the paper would greatly benefit from professional English editing. A part from the many grammar mistakes, I think that the message of the paper could be better conveyed by revising the text fixing errors and above all amending the tone and word choosing.

Thank you for these comments, the text was once again checked by the native speaker. If further corrections are needed, please provide detailed comments

Reviewer 3 Report

Thank you for inviting me to review the paper on “Sex differences in the relationship between Student School Burnout and Problematic Internet Use among Adolescents”. This is a good paper which covers two important topics: Burnout and Problematic Internet Use. The statistical analysis is methodologically sound. This paper deserves to be published by IJERPH. I have the following recommendations:

Under Introduction, the authors stated “There are several works and studies devoted to the causes, manifestations and effects of occupational burnout among these professions”. This statement needs a reference. I suggest to cite the following study published by IJERPH to support this statement.

…. occupational burnout among these professions (Low et al 2019)

Reference:

Low ZX, Yeo KA, Sharma VK et al Prevalence of Burnout in Medical and Surgical Residents: A Meta-Analysis. Int J Environ Res Public Health. 2019 Apr 26;16(9). pii: E1479.

Under Introduction, the authors stated “Additionally, characteristics of Internet-addicted students usually include alcohol and drug abuse, smoking, online and offline antisocial behavior, health problems, individuals' suicidal ideation and depressive symptoms, withdrawal and poor school performance [33, 32]. I suggest the authors to state attention deficit and hyperactivity. Please modify the statement:

depressive symptoms, withdrawal and poor school performance, as well as attention deficit and hyperactivity [33, 32, Ho et al 2014].

Reference:

Ho RC, Zhang MW, Tsang TY et al  The association between internet addiction and psychiatric co-morbidity: a meta-analysis. BMC Psychiatry. 2014 Jun 20;14:183.

Under Introduction, the authors stated “Stress was found to be a strong predictor of internet addiction [41], and internet addiction predicted stress [42]” This statement sounds repetitive. It is important to highlight that internet addiction predicted negative quality of life. Please modify the statement.

.. and Internet addiction predicted stress [42] and negative quality of life (Tran et al 2017)

Reference:

Tran BX, Huong LT, Hinh ND et al A study on the influence of internet addiction and online interpersonal influences on health-related quality of life in young Vietnamese. BMC Public Health. 2017 Jan 31;17(1):138

Authors need to be consistent to use capital letter for Internet throughout the manuscript.

Please add a space between “of” and “Internet” in the following statement:

“We cannot overlook that some of the burnout symptoms appear to resemble the ones ofInternet addiction.”

Under section 2.1, the demographic data are too brief, including age and gender. recommend the authors to include more demographic data if possible and preferably in a table. If not, please address this as a limitation (see below).

The authors stated “In summary, female high school students tend to have lessschool burnout characteristics” Please leave space between “less” and “school”.

Under limitation, the authors stated “technological addiction such as addiction from mobile phone” This needs a reference. Please quote the following study:

Carbonell X, Chamarro A, Oberst U et al  Problematic Use of the Internet and Smartphones in University Students: 2006-2017. Int J Environ Res Public Health. 2018 Mar 8;15(3).

Under limitations, the authors stated “The study did not involve the information about other risky behaviors such as smoking, alcohol or drug abuse and other internet behaviors such as social networking, sharing, banking, shopping etc or information about other technological addiction such as addiction from mobile phone.” There are quite a number of information not provided in this study. Please add the following statements.

The study did not involve the information about other risky behaviors such as smoking, alcohol or drug abuse and other internet behaviors such as social networking, sharing, banking, shopping etc or information about other technological addiction such as addiction from mobile phone. In addition, some of the common information reported by other studies on Internet addiction were not reported which could related to Internet addiction and burnout. For example, frequency of online gaming (Do et al 2019) and money spent (Rho et al 2017), information on computer ownership (Lai et al 2013), places for Internet access (Mak et al 2014),  living arrangement and parental supervision (Tran et al 2017), personality (Chang et al 2019), strategies to cope with stress (Chwaszcz et al 2018) and sleep quality (Zhang et al 2017).

References

Do HN, Onyango B, Prakash R et al  Susceptibility and perceptions of excessive internet use impact on health among Vietnamese youths. Addict Behav. 2019 Jan 31:105898

Rho MJ, Lee H, Lee TH et al  Risk Factors for Internet Gaming Disorder: Psychological Factors and Internet Gaming Characteristics.

Int J Environ Res Public Health. 2017 Dec 27;15(1).

Lai CM, Mak KK, Watanabe H, Psychometric properties of the internet addiction test in Chinese adolescents. J Pediatr Psychol. 2013 Aug;38(7):794-807.

Mak KK, Lai CM, Watanabe H et al Epidemiology of internet behaviors and addiction among adolescents in six Asian countries. Cyberpsychol Behav Soc Netw. 2014 Nov;17(11):720-8.

Tran BX, Mai HT, Nguyen LH et al  Vietnamese validation of the short version of Internet Addiction Test. Addict Behav Rep. 2017 Jul 8;6:45-50.

Chang YH, Lee YT, Hsieh S. et al Internet Interpersonal Connection Mediates the Association between Personality and Internet Addiction. Int J Environ Res Public Health. 2019 Sep 21;16(19)

Chwaszcz J, Lelonek-Kuleta B, Wiechetek M et al Personality Traits, Strategies for Coping with Stress and the Level of Internet Addiction-A Study of Polish Secondary-School Students. Int J Environ Res Public Health. 2018 May 14;15(5).

Zhang MWB, Tran BX, Huong LT et al Internet addiction and sleep quality among Vietnamese youths. Asian J Psychiatr. 2017 Aug;28:15-20.

Author Response

Thank you very much for your opinion and interest in the topic we discussed in the article.

We referred to the suggestions.

1. Thank you very much for the suggestion of the article we did not include in the introduction (Low et al.).

2. Your suggestion that IA can cost attention deficit and hyperactivity was very interesting to us, and we will include our next study on this problem.

3. We used capital letter for Internet throughout the manuscript.

We made a space between:

 „of” and „Internet” in the sentence “We can not overlook that  some of the burnout symptoms  appear to resemble the ones of Internet addiction.”

„less” and „school” in the sentence “In summary, female high school students tend to have less school burnout characteristics”

We double checked it and we are very grateful for You to catch that mistakes.

4. Improving the article, we referred to the article suggested by the reviewer (Carbonell X, Chamarro A, Oberst U et al ProblematicUse of the Internet and Smartphones in UniversityStudents: 2006-2017. Int J Environ Res Public Health. 2018 Mar 8; 15 (3) ).

5. We also agreed that it is important to refer in limitations to other forms of activity (…In addition, some of the common information reported by other studies on Internet addiction were not reported which could related to Internet addiction and burnout. For example, frequency of online gaming (Do et al 2019) and money spent (Rho et al 2017), information on computer ownership (Lai et al 2013), places for Internet access (Mak et al 2014),  living arrangement and parental supervision (Tran et al 2017), personality (Chang et al 2019), strategies to cope with stress (Chwaszcz et al 2018) and sleep quality (Zhang et al 2017).)

6. Under section 2.1, the demographic data are too brief, including age and gender. recommend the authors to include more demographic data if possible and preferably in a table.

Thank you, and we have added this as one of the limitations of our study

Round 2

Reviewer 2 Report

The Authors addressed all points and the manuscript can be accepted in the present form.